# Analysis of Students' Emotional Patterns Based on an Educational Course on Emotions Management

Inna Reddy Edara 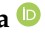

Graduate Institute of Educational Leadership and Development, Fu Jen Catholic University,
New Taipei City 24205, Taiwan; 065049@mail.fju.edu.tw

**Abstract:** Based on the current research trends and academic applications, which suggest that emotional intelligence can be learned and enhanced through education and training, this academic project coded and analyzed the emotional patterns of 46 undergraduate students who attended an 18-week academic course on emotional management. A paired-sample *t*-test showed a significant increase from pre-course to post-course emotional intelligence scores, suggesting the importance of designing an academic course in formal educational settings. Analysis of students' emotions journals indicated a total of 18 negative emotions, with the emotion of irritability occurring most frequently, followed by anger and anxiety. Loneliness, impatience, and guilt were some of the least felt negative emotions. The triggers for the arousal of negative emotions spanned family matters, interpersonal relations, academics, and communication skills, among others. The majority of the students used cognitive-behavioral and family systems theories in understanding and analyzing their emotional patterns. Students used various strategies to deal with the negative emotions, including developing communication skills, talking with family and friends, better time management, improving self-efficacy, cultivating a positive attitude, engaging in physical activities, etc. These results are discussed in this paper and the possible implications for practical purposes and further research are suggested.

**Keywords:** emotional intelligence; emotional patterns; negative emotions; responsive strategies; triggers

## 1. Introduction

Emotions are a central part of our lives [1]. In fact, emotions drive us to perform some impulsive actions. Yet, becoming aware of emotions and managing them appropriately are difficult endeavors that require skilled training in a formal academic setting. Based on this rationale, this academic-oriented project tried to analyze the specific emotions of students who attended a course on emotional management, understand their different skills of emotional management, and look for their emotional management patterns.

### 1.1. Theories of Emotions and Description of Emotional Intelligence

The main theories regarding emotions include evolutionary theories, physiological theories, cognitive theories, and appraisal theories. For example, *evolutionary theories* [2–8] recognize the adaptive function of emotions in survival mechanisms and endorse the primary emotions as directly triggered by neural activation in response to stimuli. *Physiological theories* [9,10] assume that stimuli cause a physiological activation that triggers emotions. *Cognitive theories* [11,12] believe that cognition is essential in triggering an emotion. Along the lines of cognitive theories, *cognitive appraisal* [13] proposes that the appraisal process of an event triggers an emotion, the corresponding physiological changes, and an action tendency. Thus, based on the explanation of the theoretical trends, emotions are said to be complex, for they are linked to the activation of the nervous system, physiological arousal, and behavioral tendency. Moreover, emotions are complex because they involve different

components, such as mental states and subjective experiences, cognitive processes and contextual appraisals, and expressive behaviors and cultural labels [14,15].

Closely associated with emotions is the term emotional intelligence (EI), which was made popular with the publication of "*Emotional Intelligence—Why it can matter more than IQ*" by Goleman [16]. The growth of emotional intelligence (EI) is said to be a result of two areas of research. The first is *cognition and affect*, which describes the interaction between cognitive and emotional processes to enhance thinking. The second area is *intelligence* itself, wherein scholars consider intelligence as a broader array of mental abilities [17–23].

Along these two areas of EI, scholars also developed various models to assess EI. For instance, a *trait model* that focuses on self-reporting of perceived traits and behavioral dispositions [24,25]; an *ability model* that emphasizes the individual's ability to process emotional information and adapt it to the social environment [26–29]; a *mixed model* that combines both traits and abilities [21]; a *configural model* in which the emotional states are attributed to visual and auditory nonverbal cues [30], and a *neurological model* that seeks to characterize the neural mechanisms of EI [31].

Based on the various models, emotional intelligence (EI) could be categorized as having four distinct yet related abilities: perceiving, using, understanding, and managing emotions [32]. In terms of describing EI as an array of skills, Goleman [21] delineated five skills and competencies: self-awareness, self-regulation, social skills, empathy, and motivation. Thus, based on abilities and skills, EI could be defined as the constellation of individuals' self-perceptions of their abilities and skills in becoming aware of and recognizing emotions in themselves and others, labeling them appropriately, using them to guide thinking and behavior, adapting them to suit different contexts, and achieving desired performance [33,34].

*1.2. Education to Enhance Emotional Intelligence*

As acquired knowledge and a combination of dynamic skills on how emotions function in oneself and others, EI can be learned and enhanced through education, training, and participation. Scholars [35,36] performed a critical review of the literature to evaluate whether someone could be trained in EI. Indeed, some studies suggested that EI could successfully be enhanced. For example, Srivastava and Bharamanaikar [37] suggested that an executive coaching program improved managers' levels of EI competencies. Chapman [38] indicated that a leadership coaching program specifically designed to enhance leaders' EI did in fact increase their EI. Hess and Bacigalupo [39] reported that the development of EI skills and behaviors enhanced both individual and group decisions and outcomes.

Zeidner, et al. [35] reviewed whether EI could be trained and provided some guidelines for the development, implementation, and evaluation of EI intervention programs. Their guidelines included: defining EI's conceptual framework, specifying goals and outcomes, identifying the educational context, integrating EI programs into the instructional curriculum, and generalizing the domain of EI skills.

Summarizing the literature review, it could be said that emotions are pleasant or unpleasant states of response to any given stimulus, associated with a particular pattern of physiological arousal, cognitive appraisal, and adaptive behaviors. As complex states, emotions are linked to the arousal of the nervous system, physiological activity, behavioral tendency, and motivation. As such, they involve different components, including mental processes, contextual appraisals, explicit behaviors, and cultural labels. The combination of these components is known as emotional intelligence (EI), which is described as an ability and skill of perceiving emotions in oneself and others, labeling them appropriately, using them to guide thinking and behavior, and adapting them to different life situations. As a combination of dynamic abilities and skills, EI could be enhanced through formal education and training.

*1.3. Research Design, Motivations, and Hypotheses*

Being convinced that emotional intelligence (EI) skills and competencies could be enhanced through formal education and training, this research project employed a within group, otherwise known as a one-group (a single experimental group), pre-test and post-test research design, in which a convenient target group was assessed before the intervention and after the intervention was applied. In other words, a one-group pre-test and post-test design is a pre-experimental design that applies an intervention to one group and measures the impact of the intervention, but also includes a pre-test that assesses the condition before the intervention is applied.

This pre-experimental one-group pre-intervention and post-intervention research motivations included:

1. Determining feasible academic settings for educating students about emotional intelligence.
2. Designing course content that would help students to improve their understanding of emotional intelligence.
3. Conducting a knowledge-based and skill-training academic course on emotional intelligence.
4. Encouraging students to implement knowledge and skills in managing their emotions through journal recording.
5. Analyzing the journal records for emotional patterns and assessing emotional intelligence.

In order to fulfill the above research motivations, this academic and research project involved a group of undergraduate students who attended an 18-week course on emotional management. During the course, the students were guided to write a journal of their emotions and their patterns based on the course contents and their concrete experiences, which were summarized in a written report at the end of the semester. In addition, the differences in the EI levels of the students before and after the course were evaluated.

Specifically, this research project investigated the following quantitative and qualitative hypotheses:

1. Quantitative: There would be significant increases in the pre-course scores compared to the post-course scores on the emotional intelligence (EI) of the students, after the completion of a formal academic course on EI and implementation of the appropriate EI interventions throughout the semester, indicating the importance of designing an educational program to enhance EI.
2. Qualitative: The analysis of students' emotions journals would indicate the significant patterns of negative emotions experienced, specific triggers or reasons, the application of theories and concepts to understand emotions, and the strategies employed to deal with negative emotions, emphasizing the importance of an academic course on emotional management.

## 2. Materials and Methods

### 2.1. Procedure and Participants

This research project conducted in the context of an academic course was approved by the research and curriculum committee of the researcher's institution (approval code: DOVO113873). The course was conducted as per the institutional guidelines, and the students were freely enrolled in the course and agreed to participate in the surveys. Course data were collected within an accepted educational setting to provide opportunities for participant students to learn from the instructional course. Both the quantitative and qualitative data collected were encoded and analyzed in such a manner that the identities of the participants are kept confidential.

The course on EI was designed wherein the students attended two-hour classes each week and met with the course instructor on a regular basis and as necessary to discuss their journaling of emotions, including various topics and interventions related to their emotional management (see Table 1 for the course design and the contents).

**Table 1.** Emotional Management Course Contents and Descriptive Statistics.

|  | Course Contents | Min | Max | Mean | SD |
|---|---|---|---|---|---|
| Concepts | Concepts, Definitions, Categories, Functions | 3.00 | 10.00 | 7.17 | 1.72 |
|  | Physiology and Emotions | 3.00 | 10.00 | 7.37 | 1.76 |
|  | Emotions and Cognition | 3.00 | 10.00 | 7.65 | 1.64 |
|  | Personality and Emotions | 3.00 | 10.00 | 7.11 | 1.42 |
|  | Emotions in the Context of Family | 3.00 | 10.00 | 7.67 | 1.59 |
|  | Social Culture and Emotions | 3.00 | 10.00 | 7.63 | 1.57 |
|  | Contemporary Media Influence on Emotions | 3.00 | 10.00 | 7.39 | 1.48 |
|  | Emotional Management & Technology Development | 3.00 | 10.00 | 6.89 | 1.66 |
|  | Emotions in Interpersonal Relations | 3.00 | 10.00 | 7.74 | 1.56 |
|  | Emotions and Metaphors | 3.00 | 9.00 | 6.54 | 1.66 |
| Applications | "The Angry Birds" Movie and Discussion | 3.00 | 10.00 | 7.04 | 1.44 |
|  | Analysis of Prominent Figure (Ex: Trump) | 3.00 | 10.00 | 6.46 | 1.77 |
|  | EQ Questionnaire and Analysis | 3.00 | 10.00 | 7.81 | 1.45 |
|  | Personality Test and Analysis | 3.00 | 10.00 | 7.74 | 1.68 |
|  | "Inside Out" Movie and Individual Reflection | 3.00 | 10.00 | 9.00 | 1.55 |
|  | "Inside Out" Movie, Group Discussion and Oral Report | 3.00 | 10.00 | 8.15 | 2.04 |
|  | Exercise on Developing an Emotional Vocabulary | 3.00 | 10.00 | 6.76 | 1.72 |
|  | In-vivo Exercise in Expressing Emotions | 3.00 | 10.00 | 7.28 | 1.69 |
|  | Exercise on Needs, Metaphors, and Emotions | 2.00 | 9.00 | 6.63 | 1.72 |
|  | Group Exercise on Constructing EI Patterns | 2.00 | 10.00 | 7.37 | 2.03 |
|  | Weekly Journaling and Analysis of Emotions | 3.00 | 10.00 | 7.63 | 2.03 |
|  | Paper on Personalized Emotional Management Pattern | 2.00 | 10.00 | 7.02 | 2.18 |

*N* = 46; Min = Minimum; Max = Maximum; SD = Standard Deviation.

The research sample consisted of 46 students at a prominent university in northern Taiwan, of whom 20 (43.5%) were male and 26 (56.5%) were female, with their ages ranging from 18 to 21. These participant students voluntarily enrolled in a 2-credit 18-week course on emotional management.

*2.2. Measures and Tools*

*Emotional Intelligence.* The Boston EI Questionnaire (EIQ) was used to measure EI. The EIQ was developed by Clarke [40] based on the work of Weisinger [41], and it is described as comprising self-awareness, managing emotions, self-motivation, relating to others, and emotional mentoring. The EIQ scale consists of 25 items, measured on a continuous scale of 1 to 10. Higher scores on the Boston EIQ suggest higher EI. The reliability coefficient for the total scale was 0.94 in this study. Sample items included, "Can you tell when your mood is changing" and "How well can you concentrate when you are feeling anxious?"

*Emotions Journal.* In order to analyze the significant patterns of students' emotional management, they were guided to maintain an emotions journal, regularly recording frequently experienced negative emotions, specific triggers or reasons, the application of theories and concepts to understand those emotions, and the strategies employed to deal with negative emotions. At the end of the semester, they were required to write an integral paper by tracking the journal contents and developing a specific emotional management pattern relevant to their lifestyle.

*Qualitative Content Analysis.* This qualitative design took a summative content analysis approach [42], beginning with the assessment of the frequency of the emotional vocabulary, followed by finding latent themes in terms of the participants' emotional triggers, theoretical underpinnings, and responsive strategies. This approach begins with a quantitative flavor and subsequently leads to an inductive manner of discovering codes and relevant themes. The process included defining the unit of analysis, developing a coding scheme and identifying categories, assessing coding consistency, and interpreting and drawing conclusions from the coded data [43]. In this paper, students' integral papers on emotional management patterns were analyzed for both concepts and relationships among the concepts by conducting both conceptual analysis and relational analysis. Conceptual

analysis was used to determine the existence and frequency of concepts in a given text. Relational analysis was used to examine the relationships among concepts in a given text. Coding and categorization were performed by using the "review and comment" tabs in the Word document.

## 3. Results

### 3.1. Descriptive Statistics of Course Contents

Some of the major concepts, relevant topics, appropriate interventions, and in-class or at-home assignments covered in this course are indicated in Table 1. As indicated in Table 1, the course contents included two categories: concepts and applications. Concepts included description, physiology, cognition, personality, family, society and culture, the influence of media and technology, and metaphors. Applications included watching and analyzing movies, discussing and analyzing prominent figures' emotions and management, completing relevant questionnaires, participating in in vivo exercises, writing a weekly journal, and completing a semester integration paper. The students completed the course evaluation at the end of the semester, measured on a continuous Likert-scale, with values ranging from 1 (*do not like or not satisfied at all*) to 10 (*very much like it or very satisfied*). The evaluation results indicated mean values in the range of 6.46 to 9.00. A grand mean of 7.37 suggested that the students who took the course acknowledged the importance and practicality of the content and also felt highly satisfied.

### 3.2. Differences in Pre-Course and Post-Course EI Scores

Before conducting a paired-sample *t*-test, the normality of the distribution of the sample was assessed using the Kolmogorov–Smirnov statistic, which indicated a non-significant result, $p > 0.05$, suggesting a pretty normal distribution of the sample scores on emotional intelligence. When the sample is small ($N = 46$) and with only one variable, the normality of distribution is usually not violated.

To test Hypothesis 1, the Boston EIQ was administered twice, at the beginning and end of the course. A paired-sample *t*-test was conducted to evaluate the impact of the emotional management course on the levels of participants' EI scores and the test showed a significant increase from pre-course to post-course EI scores. Specifically, there was a statistically significant increase in overall EI scores [$t$ (45) = 8.67, $p < 0.001$] from pre-course ($M = 154.39$, $SD = 37.94$) to post-course ($M = 179.37$, $SD = 33.17$). The mean increase in EI scores was 24.98, with a 95% confidence interval ranging from −30.77 to −19.17. The effect size ([44], p. 247) was very large ($\eta^2 = 0.63$; Cohen's D = 0.79), indicating the usefulness of an academic course in raising the students' EI levels. As can be seen in Figure 1 [45], the course on emotional management caused a significant increase in most of the participants' EI levels, except for participants 21, 27, and 43.

### 3.3. Analysis of Students' Emotional Patterns

The significant and relevant patterns of students' emotional management (Hypothesis 2) were studied by performing content analysis on the students' emotions journal, which included frequently experienced negative emotions, specific triggers or reasons, the application of theoretical concepts to understand those emotions, and the interventions or strategies that they used to deal with their negative emotions. The summary of the results is presented in Table 2.

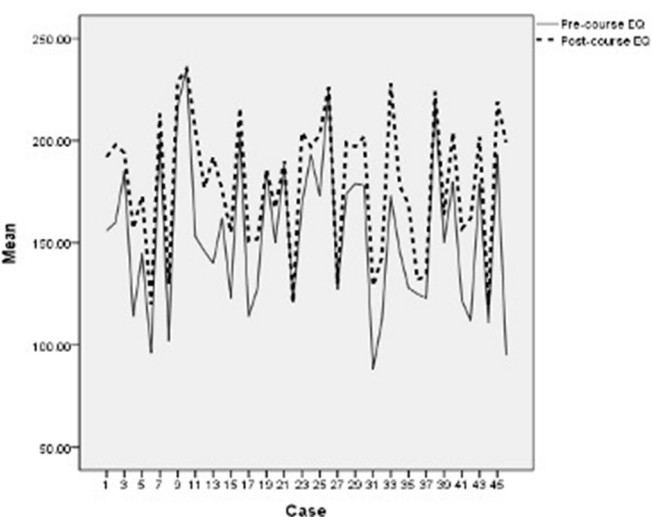

**Figure 1.** Comparison of pre-course and post-course EI (EQ) scores [45].

**Table 2.** Summary of Frequent Negative Emotions, Triggers, Theories, and Strategies.

| Emotion (Freq): P# | Trigger/Reason (Freq) | Theories | Response Strategies |
|---|---|---|---|
| 1. Irritability (14): 2, 3, 4, 10, 13, 19, 23, 24, 27, 30, 31, 42, 43, 44<br>2. Anger (13): 1, 6, 7, 9, 15, 17, 23, 30, 31, 33, 34, 37, 39, 40<br>3. Anxiety (13): 1, 4, 11, 13, 14, 19, 20, 28, 34, 39, 40, 42, 46<br>4. Sadness (8): 11, 17, 18, 29, 30, 31, 35, 45<br>5. Disappointment (5): 10, 19, 29, 33, 39<br>6. Depression (5): 1, 23, 32, 38, 43<br>7. Frustration (5): 22, 23, 26, 29, 45<br>8. Fear (4): 5, 8, 21, 39<br>9. Grief (4): 9, 15, 16, 33<br>10. Loss (2): 6, 23<br>11. Emptiness (2): 9, 10<br>12. Low self-esteem (2): 11, 36<br>13. Regret (2): 16, 43<br>14. Lonely (1): 5<br>15. Disturbed (1): 12<br>16. Shame (1): 18<br>17. Guilt (1): 25<br>18. Impatient (1): 41 | ≫ Interpersonal<br>≫ View of Success<br>≫ Academics<br>≫ Communication<br>≫ Personality<br>≫ Self-blame<br>≫ Self-value<br>≫ Other's expectations<br>≫ Family matters<br>≫ Irrational reasoning<br>≫ Self-expectations<br>≫ Blame surroundings<br>≫ Self-repression<br>≫ Pessimistic thinking<br>≫ Different values<br>≫ Different thinking styles<br>≫ Can't accept failure<br>≫ Lose face<br>≫ Others' opinion<br>≫ Perfectionism<br>≫ Self-denial<br>≫ Time management<br>≫ Unfamiliar contexts<br>≫ Self-doubt<br>≫ Polarizing<br>≫ Over sensitive<br>≫ Ambiguous<br>≫ Excessive comparison<br>≫ Future concerns<br>≫ Collaboration<br>≫ Heavy sense of mission<br>≫ Inferiority complex<br>≫ Shy nature | ≫ Ellis<br>≫ Bowen<br>≫ Lazarus<br>≫ Rotter<br>≫ Personality<br>≫ Rogers<br>≫ Maslow<br>≫ Beck<br>≫ Family of origin<br>≫ Behavioral theory | ≫ Different value perspective<br>≫ Useful Communication<br>≫ Divert attention/focus<br>≫ Talk with friends/share with family<br>≫ Detailed daily schedule<br>≫ Improve self-efficacy<br>≫ Develop can-do attitude<br>≫ Habit of maintaining emotions journal<br>≫ Reading and being active<br>≫ Think of good times<br>≫ Express appropriately<br>≫ Face them with positive attitude<br>≫ Do proper analysis and processing<br>≫ Clarify personal needs<br>≫ Deep breathing<br>≫ Emptying mind<br>≫ Acknowledge the influence of cultures<br>≫ Exercise/sports<br>≫ Share with like-minded friends<br>≫ Do not care what others think<br>≫ Affirm oneself<br>≫ Develop self-confidence<br>≫ Self-acceptance<br>≫ Movies/music/sleep<br>≫ Travelling<br>≫ Seek other's opinion<br>≫ Trace the source of emotions<br>≫ Deal objectively<br>≫ Isolation<br>≫ Adjust mentality<br>≫ Deal positively<br>≫ Engage in mental exercises<br>≫ Be responsible<br>≫ Self-exploration and observation<br>≫ Positive self-talk<br>≫ Accept the unchangeable<br>≫ Maintain rationality<br>≫ Change mind/think differently<br>≫ Avoid unnecessary comparisons |

Note: Freq = Frequency; P# = Participant Number.

As reported in Table 2, the student participants journaled a total of 18 negative emotions, with the emotion of irritability occurring most frequently, followed by anger and

anxiety. Loneliness, impatience, and guilt were some of the least felt negative emotions among the student participants.

The triggers or reasons for the arousal of negative emotions spanned family matters, interpersonal relations, academics, communication skills, personality traits, value systems, time management, thinking styles, and contextual factors. Most of the participants used cognitive-behavioral and family systems theories in understanding and analyzing their emotional patterns.

The participants' strategies in dealing with the negative emotions varied across the spectrum of developing useful communication skills, talking with family and friends, better time management, improving self-efficacy and self-confidence, cultivating a positive attitude, clarifying needs, engaging in exercise or physical activities, availing movies and music, positive self-talk, avoiding comparisons, etc.

### 3.3.1. Anxiety

Thirteen participants recorded the emotion of anxiety. Anxiety is an emotion characterized by worried thoughts, apprehensions, sensations of tension, excessive nervousness, etc. The emotion of anxiety may be caused by a mental condition, a physical condition, stressful life events, or a combination of any of these and more.

For many of the participants, the emotion of anxiety was related to academics and assignments. For example, Participant#1 said that feeling anxious about academics and homework led to sleepless nights, self-blame, and consequential bad health.

> *When working on a report, I don't get enough sleep every day, and my energy is exhausted. I often feel anxious, worried that the report will not be completed, and blame myself for being angry with myself. I ask: Why can't I do anything well? In the end, not only did I feel negative psychologically, but it also affected my health.*

To understand the emotion of anxiety, many of the participants used cognitive-related theories. For instance, Participant#1 used Albert Ellis' theory:

> *Ellis' emotion theory can be used to explain. It seems that it is not the report itself that makes me feel anxious, but my belief is to be the most perfect, can't be sloppy, and hope to be praised by others.*

In essence, Albert Ellis' theory of Rational Emotive Behavior Therapy (REBT) helps people learn to challenge their own irrational beliefs and develop the habit of thinking in beneficial and rational ways. After understanding the possible reason for anxiety as being one's own beliefs, Participant#1 was able to deal with anxiety by disputing their belief of perfectionism, learning to self-affirm and not care what others think. In Participant#1's words:

> *Everything does not need to be perfect to be applauded, as long as you do your best, you are worthy of yourself.*

### 3.3.2. Irritability

Irritability was the most frequently experienced emotion by the participants. Irritability is a state that involves feelings of frustration, impatience, and annoyance, especially over small things. Regarding the emotion of irritability, Participant#4 said:

> *I often feel irritable. I originally thought that "physiology" affects "emotion." However, I found in the emotional log that the negative emotions at this time were actually induced by other events (for example: the report was not finished, the midterm exam was approaching), and the physiological part was just a multiplication effect.*

Understanding the reasons for irritability through Rotter's theory of locus of control and engaging in positive self-talk, Participant#4 further logged:

> *And I found that I am an extremely "internal control" person, and I often have no confidence in myself. If I really don't do things well, I will blame myself for a long time.*

*Later, I tried to consciously tell myself to take it easy, and to write down the things to be done in detail. In this way, I can not only know that things are on track, but also be less likely to suffer from being irritable and screw things up.*

Participant#4 also mentioned self-efficacy, which refers to an individual's belief in one's capacity to engage in behaviors that are necessary for producing targeted performance attainments. Self-efficacy also reflects confidence in the ability to exert control over one's own motivation, behavior, and social environment. The Participant said:

*I also gradually increased my self-efficacy, believing that I can get things done well.*

### 3.3.3. Anger

Anger was also one of the prominent emotions experienced by the participants. Anger is closely related to irritability and frustration. In fact, anger is used as an umbrella term under which there are many emotions, including frustration and grief. As an umbrella term, anger is also often called a secondary emotion or a masking emotion because people tend to resort to anger in order to cover up other vulnerable feelings.

For example, Participant#6, who works a part-time job in a restaurant, chronicled issues with anger and initially thought that anger was caused by customers who make unreasonable demands.

*The negative emotion (of anger) I often experience is that when I work in a restaurant, I often encounter some customers who make unreasonable demands. My mood, after a whole day of fatigue and high-intensity stress, made my negative emotions more negative, and my inner anger also increased with the degree of fatigue.*

Later on, he realized that anger is caused by fatigue and stress, and dealt with it by chatting with colleagues. The Participant also connected unreasonably demanding customers to his family experience. In the words of this Participant:

*However, the unreasonable request of the guests is just a fuse, and how do I solve this negative emotion? Whenever I have this negative emotion, I always complain to my colleagues and chat with them to discuss it, and vent this emotion.*

### 3.3.4. Sadness

The emotional state of sadness is usually aroused by the loss of something that is dearly valued and highly cherished. Sadness is also a normal response to situations that are upsetting, painful, or disappointing.

Participant#35 journaled her emotion of sadness, usually related to relationships and her fear of being rejected. She said:

*Usually my negative emotions (of sadness) are related to relationships…I think I am a person who cares about my mirror self (image). This has been the case since I was a child. Although I have improved a lot along the way, I am worried that others will not accept me.*

Participant#35 continued to say that the course on emotional management helped her to improve her interpersonal relationships. She also narrated an incident and how she dealt with it eventually. In her own words:

*So after taking this class, I really agree that "emotions" can improve interpersonal relationships! I really wanted to participate (in Christmas dance party), but I dare not to, and I feel that it is a pity not to go… Yes, very contradictory. I just thought about why I dare not go? I think the reason is that I am afraid that I will be out of place, or not welcome there. In addition, I broke up with my boyfriend before, and I began to feel that no one would accept me, because I was a bad person, and no one would really like me. I felt that as long as I avoided the beginning of everything, I could not worry about any ending.*

Sadness caused by ending an interpersonal relationship and associated with irrational beliefs and distorted thoughts made Participant#35 decide that avoidance is the best option to deal with sadness. After going through the course, the Participant realized that avoidance was not the best option and decided to go to the party,

> *because I felt it was a pity not to go. After all, my good friends would also go, and I believe they would take care of me. After going there later, I had a lot of fun. I think it's because I adjusted my mentality, I tried to tell myself not to be so exaggerated. If everyone really doesn't like me, then why do I still have these friends by my side now! I mustered up my courage. Sure enough, the result was good.*

### 3.3.5. Depression

Depression is one of the negative emotions prevalent among the youth. The emotion of depression usually affects how people feel, think, and behave. Depression can occur for a variety of reasons, such as an upsetting or stressful life event, including bereavement, divorce, illness, job loss or financial worries, etc. In fact, different causes can often combine to trigger depression.

For example, Participant#38 often felt depressed during the night because it is during the night that people usually tend to ruminate, mostly over negative events. She said:

> *Every night, my mood tends to be generally low. It may be because I am all by myself, so it is easy to think about many things, especially now that I don't have any thoughts about the future, which makes me feel scared. At this time, negative emotions fill my heart.*

Participant#38 continued:

> *At this time (night), I have a negative attitude towards other things I am doing at the moment, and even start to hate myself and the people and things around me. I usually fall asleep with this emotion, and I am prone to insomnia, and then my body is sore the next day.*

This Participant also said that depression is a bad cycle when things are ruminated upon and negative emotions are entangled with her personality.

> *In the future, I think there may be more unsatisfactory things happening. If I keep everything in mind for too long, negative emotions will entangle me, and I will be prone to depression. This will not only affect my health, but may also affect my interpersonal relationships. There is a bad cycle. . . ..*

Participant#38 who often felt depressed, especially at night, realized that the cycle of negative emotions affects physical health and interpersonal relationships. This Participant also became aware of good emotional management skills. She said:

> *After taking this emotional management course, I have a deeper understanding of my emotions, and then I will work hard to control my emotions. Whenever something unpleasant happens, I will take a deep breath first, think about why this happened, why I am unhappy, is it correct for me to be unhappy. . . etc., calm myself down, and don't let emotions control (me). . ..*

### 3.3.6. Fear

Fear is one of the most basic and primal human emotions that involves a universal biochemical response and an individual emotional response. Fear alerts us to the presence of danger or the threat of harm. From infancy, people are equipped with the survival instincts necessary to respond with fear when they sense danger, whether that danger is physical or psychological.

For example, facing natural disasters is a fearful thing that might disturb normal life. Participant#8 shared:

*During one of the typhoon days, the school was closed, and this day happened to be my part-time job, so I went to 711 with a feeling of fear and reluctance. I think that going to work on a typhoon is a very dangerous thing.*

Participant#8 dealt with the emotion of fear by using Lazarus' cognitive appraisal theory:

*Although I really didn't want to go to work, I boarded the bus and I started to apply Lazarus' cognitive evaluation theory. I thought it was negative in the primary evaluation, and then entered the secondary evaluation. I thought in my heart that there would be no guests coming during the typhoon, so I chose to accept it. After a series of (evaluative) thoughts, my mood changed.*

Lazarus' theory of cognitive appraisal is an assessment of an emotional situation wherein individuals evaluate how the event will affect them, interpret the various aspects of the event, and arrive at a response based on that interpretation. Participant#8 re-interpreted the typhoon situation as not as dangerous as he initially evaluated, re-imagined that he would not be as busy as he thought for customers might not come out in a typhoon, and thus decided to go to work with a changed mood.

### 3.3.7. Guilt

Guilt is a self-conscious emotion that originates from a process of introspection and self-evaluation about one's potential responsibility for or role in an unacceptable outcome. It is also usually felt when one fails to meet particular standards.

For instance, Participant#25 shared a feeling of guilt for not being punctual to work due to various valid reasons such as attending classes and her manager thinking that she deliberately comes late. Participant#25 said:

*When I got to the part-time job, I could only apologize to other colleagues, because my "intentional" lateness caused them a lot of burden. Even though everyone told me not to take it seriously, I still felt an inexplicable guilt in my heart, partly because of being hurt (by my boss).*

This Participant used Beck's emotional cognition theory to analyze this incident.

*I found that at the moment when the incident happened, I only thought of the bad. I felt that it was a very bad thing to be rushed even though it was not yet time for work. Then I blamed myself for why I agreed to such an unreasonable scheduling of work. In addition to being guilty, I could only blame myself constantly, resulting in a very uncomfortable state of mind and body.*

Participant#25 further stated:

*I think it was a matter of lack of self-awareness that caused guilt at that time, so if this happens again, I will choose to face my boss in a rational way, and I will express my feelings to him in a tactful way, instead of arguing with him.*

### 3.3.8. Grief and Loss

Grief is the natural human reaction to loss. Some examples of loss include the death of a loved one, the ending of an important relationship, the loss of a job, loss through theft, or the loss of independence through disability.

Participant#16 talked about grief at the death of his grandfather. He said:

*My grandfather passed away recently. Because of the deep relationship with my grandfather, I was actually quite shocked and sad. In that mood, I recalled the memories with my grandfather. After a period of time, my heart calmed down, and I thought about it from another angle, and my mood was more positive. And on the day of the funeral, although my emotions were still sad, they were much more relaxed than before.*

The grieving processes that he went through helped him come to terms with the loss and transformed his view of death:

*I have slightly changed my view of death. If a person has been satisfied in this life, he also feels that he has no regrets, even in facing death. As for the death of my grandfather, it took me a while to fully recover. I diverted my attention by listening to music, exercising, and focusing on my assignments and activities.*

Participant#9 talked about his loss of friendship and companionship with a girl. He thinks that the relationship between the two sexes seems to be particularly important in life now, but it is also particularly awkward. He recalled:

*Caring about her every move, maybe nothing to her, made me laugh and cry. For a few days, I tried to divert my attention. When I was too busy with schoolwork and other affairs, I felt even more empty. Maybe I didn't have a good relationship with others. Besides the relationship between work and chatting with her, I seem to seldom chat with other friends. That's why once I don't maintain a relationship with her, I find that I really have nothing.*

How to deal with this emotion is a struggle that Participant#9 seems to have experienced. He said:

*Taking a deep breath, clearing your head, and doing other things, which is what most people do, but unfortunately it doesn't seem to work (for me).*

*The message is placed there (in my mind), and there are countless emotions in my head, I can't stop thinking about it. Even after turning it off and doing other things, the message still lingers in my mind. The seemingly innocuous things, maybe because of her, make me very sensitive.*

Finally, in dealing with his emotion of loss of friendship, Participant#9 seems to have figured out the way to deal with it. He said:

*(May be) I don't have to care what happened before. In the course of emotional management, I learned various theories and emotion-related research, and I also learned how to face up to my emotions and quickly discover my negative emotions. Be able to think about how to deal with it immediately after the emotional trigger and find out the ways before it will hurt other people, I think I am very proud to be able to check what attitude I use to face such a world and life.*

In summary, as reported in Table 2 and briefly interpreted in the aforementioned paragraphs, students who attended the course on emotional management experienced a total of 18 negative emotions, with some emotions occurring more frequently than others. Further, this variety of negative emotions was triggered by academic requirements, interpersonal relations, communication skills, personality traits, cognitive factors, and contextual situations. On the final analysis of students' integration journaling, it was obvious that they used various approaches in understanding and analyzing their emotional patterns, including cognitive-behavioral approaches, family and interpersonal systems, and locus of control theories. Students' strategies in dealing with the negative emotions included developing useful communication skills, venting to family and friends, improving self-efficacy and self-confidence, cultivating a positive attitude, reappraising cognitions and restructuring beliefs, and engaging in physical activities, among others.

## 4. Discussion

In summary, emotional intelligence (EI) is knowledge and a set of skills of how emotions function in oneself and others in a given context. Current theoretical trends suggest that EI is a combination of dynamic skills and competencies that can be learned and enhanced through education, training, and participation. Given these assumptions, this research study designed an 18-week academic and practical course on emotional management for a class of undergraduate students, investigated the impact of a formal course on the students' EI levels, and assessed the students' patterns of emotional management by analyzing their emotions journal. Appropriate statistical and qualitative analyses sug-

gested significant results and emotional management patterns, which are discussed in the following paragraphs.

*4.1. Role of the Emotional Management Course in Enhancing EI Levels*

Emotional intelligence (EI) is described as both knowledge of emotions and awareness of how they function in oneself and others. As a combination of dynamic skills and competencies, current research trends and academic applications suggest that EI can be learned and enhanced through education and training [45,46]. As the results of paired-sample *t*-tests suggested, attending a well-designed and structured emotional management course made a significant difference in the levels of students' EI levels. The mean increase in EI scores was 24.98 with a large effect size, suggesting the practicality of designing a course on emotional management.

As per the recommendations of Zeidner, et al. [35], the EI scores of the study participants were enhanced because the structured course on EI enhancement was conducted as a part of the instructional curriculum of an undergrad program. The course was also designed in such a way that a clear and integral conceptual framework was structured that included domains of physiology, cognition, personality, affect, motivation, family, culture and society, and media and technology. This integral framework conforms to the scholarly recommendations that emotions involve different components, such as contextual appraisals, cognitive processes, expressive behaviors, and socio-cultural labels [15].

When designing and instructing a course on emotional management, Zeidner, et al. [35] suggested specifying the goals of EI courses and outlining the expected outcomes of EI enhancement programs by identifying the educational purposes of the participants, the developmental context for EI interventions, and finally generalizing the domain of EI skills. In this research, these scholarly recommendations were implemented not only through conceptual descriptions but also via practical tasks, such as group discussions and oral reports, in-class group activities and exercises, weekly journal writing, and a final integration paper. As suggested by the study results, all of these content areas and skill domains must have played a significant role in enhancing the students' EI levels.

*4.2. Patterns of Emotional Management*

As an important driver of cognitive activity and individual behavior, the formation of personality, and the handling of interpersonal relationships, emotion has an important impact on the overall well-being of college students. The emotion behind cognition and behavior is not only an expression of the results of the behavior, but also represents some kind of adaptive motivational factor. Therefore, it is very important for college students to manage their emotions effectively, particularly negative emotions.

Negative emotions are described as those feelings which cause individuals to be miserable and sad. Negative emotions make people dislike themselves and others, reduce confidence and self-esteem, and reduce general wellness. Negative emotions can dampen one's enthusiasm for life, depending on the length and intensity of their impact on us and also the way we choose to express them.

Given the prevalence of negative emotions among college students, training in emotion regulation or management is deemed necessary. Emotion regulation involves (a) awareness and understanding of emotions, (b) acceptance of emotions, (c) ability to control impulsive behaviors, and (d) developing flexible strategies to modulate emotional responses in order to meet individual goals and situational demands [47]. Hence, this project designed an emotional management course that somehow fulfilled the dimensions of emotion regulation as prescribed by various scholars [15,33,46,47].

4.2.1. Pool of Negative Emotions

Both the quantitative and qualitative investigation indicated the significance of the well-structured course and the subsequent analysis of the students' journaling of their

emotions. Studies have shown that negative emotions and mental health problems are prevalent among today's college students [35,46,48,49].

As suggested by this research project, university or college students face so many demands and stressful situations that many naturally report often feeling irritable, anxious, angry, disappointed, depressed, sad, empty, or lonely. University students often seek counseling for anxiety, depression, frustration, sadness, academic performance, family issues, and relationship problems.

Negative emotions have many adverse consequences. They stop people from thinking rationally, behaving appropriately, and seeing situations in their true perspective. When this occurs, individuals tend to see only what they want to see and remember only what they want to remember, which prevents people from fulfilling their goals. The longer this process is prolonged, the more complicated the problems become. Therefore, learning to skillfully deal with negative emotions is very important.

### 4.2.2. Triggers and Reasons

In their journals, students recorded various reasons for their negative emotions. They comprised academics, family, interpersonal, personality, cognitive, cultural, future concerns, etc. For example, the family's role in the development or triggering of emotion can be seen in parenting practices, emotional family climates, family values and expectations, and different emotional learning experiences. Unhealthy interpersonal relations and a lack of communication skills also trigger negative emotions. Personality traits also play an important role in triggering specific emotions. Studies have shown a significant relationship between personality and emotions, specifically how personality traits influence the type of emotions that predominate people's lives. Further, cultural values and expectations have a role in triggering certain emotions. For example, the cultural value of 'saving face,' which means to keep others from losing respect for oneself or to avoid embarrassment, is highly valued in Oriental cultures, and doing anything that leads to 'losing face' may trigger a gamut of negative emotions.

In addition to pursuing academic excellence, the important things that are on students' minds when they start college are the new adventures and new people that will be met on the journey. Being immersed in a new environment, students can experience certain levels of stress and the consequent negative emotions that block their ability to socialize and reach their academic goals.

College students confront many challenges in pursuit of their educational goals and career paths. When such challenges and experiences are perceived as negative or when the students lack regulation mechanisms, such challenges and experiences can have an adverse effect on students' motivation, performance, and interpersonal endeavors.

### 4.2.3. Theoretical Concepts

As the journaling exercise indicated, students in this research project not only became aware of their various emotions and the potential triggers but also were able to use some of the theoretical concepts to analyze and understand their emotions. This is an important dimension of emotion regulation [15,47,50]. Without understanding emotions through proven theoretical concepts, the effective management of emotions may not be achieved.

The results of this study suggested that most of the students used cognitive-behavioral, needs, control, and personality theories and concepts to understand and analyze their negative emotions. The cognitive-behavioral approach is a form of analysis and therapy that can be used to treat people with a wide range of mental health problems. It is based on the idea that our cognition (how we think), emotion (how we feel), and behavior or action (how we act) all interact together. Specifically, our thoughts determine our feelings and our feelings lead to our behavior. Therefore, negative or unrealistic and distorted thoughts can lead to distress or other negative emotions and result in problems [33,46,51,52].

Personality itself also seems to explain how people differ in their experiences of emotions [53]. With regard to understanding emotions through personality theories, re-

search has identified personality traits that are able to explain how people behave throughout their lifespan and the way individuals deal with negative emotions. Studies also focused on individual differences in responsiveness to negative emotions, of which personality and emotion regulation strategies are said to be imperative. Indeed, individual differences were found in the habitual and spontaneous use of emotion regulation strategies in managing emotions [54,55].

The needs theory says that if the desire for change is based on an unmet need, this needs to be analyzed. Based on a concrete situation in which a particular need is strong, the feeling in this situation and its trigger are identified, analyzed, and dealt with. Possibilities of change and obstacles must be clarified to finally reach the decision of a new way of dealing with the need [56,57]. For example, the emotion of frustration is wanting something and not getting what we need. Disappointment is the feeling that there is something you want in your life that you may never get.

Another theoretical concept that was employed by the study participants was the control theory. Locus of control (LOC) was formally introduced and defined as a psychological construct by Rotter [58,59]. This theory posits that individuals with internal control expectancies will perceive events as contingent upon their own relatively permanent characteristics. In contrast, individuals with external control expectancies are more likely to perceive an event as not to be entirely contingent upon their own actions but rather as a result of luck, chance, fate, or unpredictable forces surrounding them. Rotter also cautioned researchers not to assume internal control expectancies always to be "good" and external control expectancies always to be "bad" in and of themselves, but to also consider the relation between LOC and the given situation.

### 4.2.4. Intervention Strategies

Corresponding to the triggers of negative emotions and the use of different theoretical concepts, the participants also employed various response strategies and interventions in dealing with their negative emotions. These response strategies included cognitive, behavioral, communicative, affirmative, familial, value-based, perspective-taking, expressive, analytical and processive, physical, and many others.

Specifically, cognitive responses included analyzing and processing, acceptance, positive thinking, refocusing on planning, putting into perspective, and positive self-talk and reappraisal. Behavioral strategies included physical exercise, deep breathing, emptying the mind, engaging oneself in alternative activities, watching movies or listening to music, isolating oneself from the negative emotional scene or context, reading, etc. Relational and communicative intervention strategies included developing communication skills, talking with friends, sharing with family, taking others' opinions, sharing with like-minded people, and avoiding comparison with others.

Overall, the subjects in this research appeared to have developed specific response strategies in relation to their respective negative emotions and thus exhibit the ability to effectively exert control over their emotions. As growing adults, these students know that they are expected to manage their emotions in ways that are culturally appropriate and socially acceptable, and thus successfully navigate their lives. To this end, the course on emotional management not only helped the participants to trace their emotional regulation patterns but also elevated their EI skills.

### 5. Implications and Limitations

Given that we all go through emotions and there is a necessity to manage them properly, it is natural to ask whether emotional management skills and competencies can be learned or even enhanced in order to live an emotionally positive and healthy life. The significant results from this study suggest that EI knowledge, skills, and competencies can indeed be learned or even enhanced through formal education and training that integrates conceptual knowledge and practical management skills. In other words, the study findings indicate that all of the EI domains are both teachable and learnable, which can be done

by relatively simple didactic methods that correspond to the participants' developmental stages and needs. The teachable and learnable EI courses also can be designed over a relatively short period of time in any sort of setting. The results from this study show that formal and structured programs in EI can not only make a significant difference in emotional management, but also support the idea that valid and reliable EI measures can effectively be used to assess the progress achieved as a result of the EI enhancement courses.

Future research projects involving EI domains should engage in a more extensive investigation of a variety of both instructional and applicational parameters, so that the extent to which positive changes account for a given period of time could be efficiently investigated and evaluated. The significant results presented in this research paper, along with future research projects involving more EI domains and different study participants, may have a chance to enter not only established institutions and formal education but also informal training programs, social organizations and communities, healthcare settings, and even family environments. Leaders in various settings, educators in different domains, professionals in a multitude of health arenas, and primary caregivers in family environments could benefit from training in EI so that their EI is enhanced, and hopefully this will have a positive effect on their personal lives and the surrounding ecology.

Lastly, this particular research project is not without limitations. The sample size in this study, although large enough for qualitative analysis, is not enough for quantitative analysis, particularly for estimating the effect size of pre-course and post-course EI levels, thus restricting the generalizability of the results. Future studies could be conducted with larger sample sizes and involve different cultural samples to replicate the results of this study. Further, future studies should include non-academic or informal settings with populations at different biological and psychosocial developmental stages, based on which more appropriate and relevant EI enhancement programs or courses could be designed. Such endeavors may help to expand the research in EI understanding and enhancement, and consequently, the enhanced EI may create a positive impact on various life domains of the people.

**Funding:** This research received no external funding.

**Institutional Review Board Statement:** This academic research was conducted as per the guidelines of the Declaration of Helsinki, and the course was approved by the research and curriculum committee of the researcher's institution (approval code: DOVO113873). Course was conducted and data were collected within an accepted educational setting to provide opportunities for participant students to learn from the instructional course. The data collected were encoded and analyzed in such a manner that the identity of the participants cannot readily and directly be ascertained.

**Informed Consent Statement:** Informed consent was obtained from all subjects involved in the study.

**Data Availability Statement:** Data and reports for the current study are available upon request.

**Conflicts of Interest:** The author declares no conflict of interest.

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
