# Peer review of "Analysis of Students’ Emotional Patterns Based on an Educational Course on Emotions Management"

_education, doi:10.3390/educsci13070757_

Round 1
Reviewer 1 Report (Previous Reviewer 1)
The authors have appropriately made the proposed changes in most cases. However, they have not responded satisfactorily to indications 4, 5, 7 and 12. Therefore, they must perform:
4. It is not a matter of including references to quantitative studies in their theoretical framework, but of updating some authors. Normally, a good theoretical framework should contain fundamental references (yours has them) and other current ones that address the state of this issue.
5. The authors can indicate the type of hypothesis. Therefore, please do so.
7. Indeed, if the authors have compared a group before and after an intervention, their design is "pre-experimental". This is independent of whether the data collection techniques are qualitative or quantitative. Authors are requested to review the types of designs and make it clear.
12. Science changes, and although the references are old but valid, the scientific method requires using current studies to contrast their findings. For example, if I invent a new type of computer, I could say that it is very fast compared to one from 15 years ago. However, the differences would be much smaller if I compare it with another from 5 years ago at most. Authors are requested to make an effort to include some more current references.
Author Response
Reviewer Comments and Suggestions
The authors have appropriately made the proposed changes in most cases. However, they have not responded satisfactorily to indications 4, 5, 7 and 12. Therefore, they must perform:
- It is not a matter of including references to quantitative studies in their theoretical framework, but of updating some authors. Normally, a good theoretical framework should contain fundamental references (yours has them) and other current ones that address the state of this issue.
- The authors can indicate the type of hypothesis. Therefore, please do so.
- Indeed, if the authors have compared a group before and after an intervention, their design is "pre-experimental". This is independent of whether the data collection techniques are qualitative or quantitative. Authors are requested to review the types of designs and make it clear.
- Science changes, and although the references are old but valid, the scientific method requires using current studies to contrast their findings. For example, if I invent a new type of computer, I could say that it is very fast compared to one from 15 years ago. However, the differences would be much smaller if I compare it with another from 5 years ago at most. Authors are requested to make an effort to include some more current references.
Response to Comments 4 and 12
Thanks to the reviewer for suggesting to include some recent references. I have tried to include as many as possible, such as the following:
- Cherry, K. The 6 major theories of emotion. Verywell Mind, Retrieved from https://www.verywellmind.com/theories-of-emotion-2795717
- Barrett, L.F.; Satpute, A.B. Historical pitfalls and new directions in the neuroscience of emotion. Neuroscience Letters, 2019, 693, 9–18.
- Al-Shawaf, L.; Conroy-Beam, D.; Asao, K.; Buss, D.M. Human emotions: An evolutionary psychological perspective. Emotion Review, 2016, 8(2), 173–186.
- Krishanu, K. D. A Study on Evolutionary Perspectives of ‘Emotions’ and ‘Mood’ on Biological Evolutionary Platform. Psychology and Behavioral Sciences, 2018, 7(5), 89-96.
- Perry, N. B., & Calkins, S. D. A biopsychosocial perspective on the development of emotion regulation across childhood. Emotion Regulation, 2018, 3-30.
- Smagorinsky, P. (2021). The relation between emotion and intellect: Which governs which? Integrative Psychological and Behavioral Science, 2021, 55(4), 769–778.
- Cong-Lem, N. Emotion and its relation to cognition from Vygotsky’s perspective. European Journal of Psychology of Education, 2023, 38, 865–880.
- Petrides, K.V., & Mavroveli, S. Theory and applications of trait emotional intelligence. Psychology, 2018, 23(1), 24-36.
- Pérez-González J-C, Saklofske DH and Mavroveli S. Editorial: Trait emotional intelligence: Foundations, assessment, and education. Frontiers in Psychology, 2020, 11:608
- Mayer, J.D., Caruso, D. R., & Salovey, P. The ability model of emotional intelligence: Principles and updates. Emotion Review, 2016, 8, 1-11.
- Fiori, M., Vesely-Maillefer, A.K. Emotional Intelligence as an ability: Theory, challenges, and new directions. In: Keefer, K., Parker, J., Saklofske, D. (eds) Emotional intelligence in education: The Springer series on human exceptionality, 2019. Springer, Cham. https://doi.org/10.1007/978-3-319-90633-1_17
- Elfenbein, H.A., & MacCann, C. A closer look at ability emotional intelligence (EI): What are its component parts, and how do they relate to each other? Social and Personality Psychology Compass, 2017, 11.
- Maddocks, J. Introducing an attitude-based approach to emotional intelligence. Frontiers in Psychology, 2023, 13.
- Bucich, M., & MacCann, C. Emotional intelligence research in Australia: Past contributions and future directions. Australian Journal of Psychology, 2019, 71, 59–67.
- MacCann, C., Jiang, Y., Brown, L. E., Double, K. S., Bucich, M., & Minbashian, A. Emotional intelligence predicts academic performance: A meta-analysis. Psychological Bulletin, 2020, 146(2), 150.
- Hughes, D.J., Kratsiotis, I.K., Niven, K. Personality traits and emotion regulation: A targeted review and recommendations. Emotion, 2020, 20 (1), 63-67.
Response to Comment 5
The type of hypothesis is indicated in Section 1.3
Response to Comment 7
Research design is added and explained in Section 1.3
Reviewer 2 Report (Previous Reviewer 2)
This article is a major improvement. Great job. I ask for a major revision, but both requests are minor and require little time.
1. Reference how you conducted content analysis. Connect to traditions in content analysis. Perhaps thematic analysis?
2. You provide themes, but you have no introductory paragraph to explain each one (4-6 sentences); there could be a concluding paragraph as well). Please include an introductory paragraph to explicate the who-what-where-when-how-why of themes. What is the theme? How defined? What does the theme not mean? How does it work? Otherwise, you have somany examples, you do not tie all the examples together.
Though not required, you could list the examples which either do not fit or challenge your assumptions.
Language seems fine/proofread again.
Author Response
Reviewer Comments and Suggestions
This article is a major improvement. Great job. I ask for a minor revision, but both requests are minor and require little time.
- Reference how you conducted the content analysis. Connect to traditions in content analysis. Perhaps thematic analysis?
- You provide themes, but you have no introductory paragraph to explain each one (4-6 sentences); there could be a concluding paragraph as well). Please include an introductory paragraph to explicate the who-what-where-when-how-why of themes. What is the theme? How defined? What does the theme not mean? How does it work? Otherwise, you have so many examples, you do not tie all the examples together.
Response to Comment 1
The following description of qualitative content analysis is added in Section 2.2.
This qualitative design took a summative content analysis approach [45], beginning with the assessment of the frequency of the emotional vocabulary, followed by finding latent themes in terms of the participants’ emotional triggers, theoretical underpinnings, and responsive strategies. This approach begins with a quantitative flavor and subsequently leads to an inductive manner of discovering codes and relevant themes. The process included defining the unit of analysis, developing a coding scheme and identifying categories, assessing coding consistency, and interpreting and drawing conclusions from the coded data [46].
Response to Comment 2
As the reviewer suggested, wherever it was felt necessary, I have included a brief introduction to the thematic emotions in Section 5.3 and the sub-sections.
Round 2
Reviewer 2 Report (Previous Reviewer 2)
Much improved from first submission.
This manuscript is a resubmission of an earlier submission. The following is a list of the peer review reports and author responses from that submission.
Round 1
Reviewer 1 Report
Dear editor of the journal “Education Sciences”,
Thank you very much for the opportunity to review the manuscript entitled “Making Sense of Emotions Management Course: Analysis of Emotions Journal”.
In general, the authors present a manuscript of some interest, but that presents serious problems for it to be published. Below are several indications that are essential for this purpose (I hope they will be helpful to the authors):
- The title is confusing. It seems that a review of the scientific production of a magazine called “Emotions Journal” has been carried out, when in fact a pre-experimental study has been carried out.
- The proposed topic is exclusively associated with the field of psychology, so it does not respond very well to the focus of this journal. These variables "emotional intelligence" should be related to academic factors so that it is appropriately adapted to its objective.
- The summary can be greatly improved: the sample, design or instrument used is not indicated. It is recommended to make it shorter (because it is a summary), since it has 320 words! For this, the authors should focus on the objective, the methodology and the main findings.
- The theoretical foundation is more or less appropriate. However, it presents a big problem: while it is appropriate to rely on key authors (for example, Daniel Goleman), 95% of the references are not up to date. As an example, I put their years in order: 1872, 1992, 2006, 1980, 1884, 1960, 1966, 1962, 1986, 1988, 1984, 2015, 1995, 1995... It is not, until reference number 24, that the first appointment of the last 5 years appears, and the trend continues with very old references. It would be essential to carry out an in-depth review of the last 5-10 years, and modify the theoretical framework, since several of these concepts have evolved.
- The hypotheses are well developed, although their type is not indicated (unilateral, bilateral, null...). In addition, they do not contribute anything new: obviously, the subjects who follow the program will improve their emotional intelligence. The authors do not develop a research problem either.
- Material and method: the design is not indicated (at a quantitative level, it seems that it is a pre-experimental study without an equivalent control group). On the other hand, it would fit into a mixed design when carrying out the analysis of the magazine that the participants have prepared.
- There is no verification of the normality of the sample to use T-test. Kolmogorov-Smirnov should be used to see normality, and if it does not exist, use the Wilcoxon test.
- The analysis of the changes given in the evaluation with the Boston EI Questionnaire is very flat, since it is carried out for the 25 items. Ideally, it would be to calculate the dimensions used by the questionnaire, in order to determine the changes given in them.
- The procedure is unclear: the details of the course are not detailed at the educational level (something essential) nor are ethical issues (for example, whether the study obtained the approval of the ethics committee of your institution).
- Regarding the results, the qualitative analysis is very flat. Some software could have been used for this purpose, such as NVIVO, MAXQDA, etc. Or, at least, a matrix of interrelation of the concepts and the participants could have been elaborated.
- In my opinion, the discussion is not appropriate. It is well founded, but a discussion has to use current studies. The first quote used in the discussion (number 36), from now on, are from the following years: 2013, 2004, 2015, 2002, 2004, 2015, 2004, 2013, 2004, 2002,... Many are almost 20 years old. , and the most current ones do not reach the last 5. The authors should make an effort to review the literature from 2017 onwards.
For all these reasons, my opinion is that this manuscript should be rejected. I'm sorry for this disappointing news. I hope the authors can improve it.
Reviewer 2 Report
Why is table 1 in the procedures? Besides mentioning the course evaluation, how does so much information contribute to the study? What is the validity/reliability of the course evaluation, how was it developed, and did everyone complete it?
What computer program did you use?
Why did you use eta squared with the t-test? That is not a common procedure. Eta squared can be biased, so I think Cohen's D might be better.
You cannot necessarily state the course was the cause, as there might be other reasons. You can infer, but there could be other factors.
You analyze the journals, but you fail to mention in the methodology how this is done. Do you just complete a keyword search?
You seem to randomly pick large quotes without explanation. You had a sample of 46, so you should give brief parts across the sample. You do not analyze and bring up key concepts, minor concepts, and data which do not fit.
Also, how did you protect anonymity and confidentiality? Finally, there should be multiple thoughts and ideas which do not fit.
Qualitative research demands you establish validity and reliability.
English language is fine.